# Inhibitors of Ceramide- and Sphingosine-Metabolizing Enzymes as Sensitizers in Radiotherapy and Chemotherapy for Head and Neck Squamous Cell Carcinoma

**DOI:** 10.3390/cancers12082062

**Published:** 2020-07-26

**Authors:** Yoshiaki Yura, Atsushi Masui, Masakazu Hamada

**Affiliations:** 1Department of Oral and Maxillofacial Surgery, Osaka University Graduate School of Dentistry, Suita, Osaka 565-0871, Japan; hmdmskz@dent.osaka-u.ac.jp; 2Department of Oral and Maxillofacial Surgery, Itami City Hospital, Itami, Hyogo 664-8540, Japan; amasui@hosp.itami.hyogo.jp

**Keywords:** head and neck squamous cell carcinoma, radiotherapy, chemotherapy, ceramide-metabolizing enzyme, enzyme inhibitor, molecular targets

## Abstract

In the treatment of advanced head and neck squamous cell carcinoma (HNSCC), including oral SCC, radiotherapy is a commonly performed therapeutic modality. The combined use of radiotherapy with chemotherapy improves therapeutic effects, but it also increases adverse events. Ceramide, a central molecule in sphingolipid metabolism and signaling pathways, mediates antiproliferative responses, and its level increases in response to radiotherapy and chemotherapy. However, when ceramide is metabolized, prosurvival factors, such as sphingosine-1-phosphate (S1P), ceramide-1-phosphate (C1P), and glucosylceramide, are produced, reducing the antitumor effects of ceramide. The activities of ceramide- and sphingosine-metabolizing enzymes are also associated with radio- and chemo-resistance. Ceramide analogs and low molecular-weight compounds targeting these enzymes exert anticancer effects. Synthetic ceramides and a therapeutic approach using ultrasound have also been developed. Inhibitors of ceramide- and sphingosine-metabolizing enzymes and synthetic ceramides can function as sensitizers of radiotherapy and chemotherapy for HNSCC.

## 1. Introduction

Head and neck cancers are malignant tumors of the oral cavity, pharynx, and larynx, with the sixth highest incidence worldwide, accounting for 5% of all cancers [1,2,3]. Over 90% of head and neck cancers are squamous cell carcinomas (SCC), and the majority of oral cancers are also SCC [4]. Surgery is prioritized for the treatment of oral SCC. Among digestive organ cancers, a characteristic of head and neck SCC (HNSCC), including oral SCC, is the availability of radiotherapy. For advanced oral SCC, postoperative ionizing radiation with X-rays is usually combined with chemotherapy to improve the outcomes. The US Food and Drug Administration (FDA)-approved chemotherapeutic agents for HNSCC include cisplatin, methotrexate, 5-FU, bleomycin, docetaxel, and small-molecule targeted drugs [5,6]. Combination of radiotherapy with chemotherapy, when overdosed in any combination, can lead to serious acute side effects and late-onset dysfunction [7]. Therefore, safe and effective sensitizing drugs against radiotherapy and chemotherapy are needed.

The Cancer Genome Atlas (TCGA) program reported the results of whole-genome sequencing on tumor tissues from 279 patients with HNSCC [8]. This led to the division of HNSCC into two subsets with different clinical and molecular profiles: human papillomavirus (HPV)-positive tumors, which generally have a better prognosis, and HPV-negative tumors with a poorer prognosis. Although HPV-associated oropharyngeal cancer has a better prognosis, evaluation of HPV and p16 expression in oral SCC failed to demonstrate any survival benefit [9]. In HPV-negative HNSCC, activating mutations in classic oncogenes are relatively rare, and most genetic alterations are in tumor suppressor genes such as p53 and cyclin-dependent kinase inhibitor 2A (CDKN2A). This finding is important to develop therapeutic agents because the discovering of new compounds to restore the activity of altered tumor suppressor genes is highly challenging [10]. On the other hand, activation of numerous signaling pathways has been implicated in cell survival, proliferation, angiogenesis, and/or inflammation, and many molecules in theses pathways have emerged as potential targets [11,12]. For HNSCC, monoclonal antibodies against epidermal growth factor receptor (EGFR) and human EGF receptor 3 (HER3), in addition to as small-molecule inhibitors against serine threonine kinase, cyclin-dependent kinase and tyrosine kinase are under clinical investigation [13,14,15]. Other types of approaches are also required.

Among sphingolipids constituting cell membranes, ceramide, a central molecule in sphingolipid metabolism and signaling pathways, is a typical sphingolipid exhibiting cytotoxicity [16,17,18,19,20,21]. In cancer cells, its levels and signaling are usually suppressed by the overexpression of ceramide-metabolizing enzymes or down-regulation of ceramide-generating enzymes. Radiotherapy and chemotherapeutic drugs increase intracellular ceramide levels, and this increase can restore the therapeutic sensitivity of HNSCC to these therapies [22,23,24,25]. Therefore, drugs that inhibit ceramide-metabolizing enzymes are expected to be sensitizers of radiotherapy and chemotherapy. In this article, we review the metabolic pathways of ceramide, its function, the inhibitors of ceramide- and sphingosine-metabolizing enzymes, and their mechanisms of action.

## 2. The Pathways of Generation and Metabolization of Ceramide

Sphingolipids are structural molecules of cell membranes and important regulators of proliferation, migration, invasion, and metastasis of cancer cells. Ceramide can be generated through the hydrolysis of complex sphingolipids, de novo synthesis pathway, or salvage pathway [26]. In the sphingomyelinase (SMase) pathway, SMase hydrolyzes plasma membrane sphingomyelin, and produces ceramide and phosphocholine [27,28,29]. There are acidic, neutral, and basic isoforms of SMase. The de novo synthesis of ceramide begins in the endoplasmic reticulum (ER) with the condensation of serine and palmitoyl-CoA by serine palmitoyl transferase (SPT) to form 3-ketodihydrosphingosine (3-ketosphinganine), which is, then, reduced to dihydrosphingosine (sphinganine) and acylated by dihydroceramide synthase, referred to as ceramide synthase (CerS), to produce dihydroceramide. Lastly, the desaturation of dihydroceramide by dihydroceramide desaturase (DES) generates ceramide (Figure 1). In the salvage pathway, ceramides can be synthesized, re-utilizing free sphingosine formed by the degradation of glycosphingolipids and other complex sphingolipids [30].

Ceramide is metabolized via the action of many metabolizing enzymes. Three classes of ceramidases (CDases)—acid, neutral, and alkaline—have been identified and distinguished by their pH for optimal activity [31,32,33]. CDase hydrolyzes ceramide to yield sphingosine, which is phosphorylated by sphingosine kinase 1 (SphK1) and SphK2 to generate sphingosine-1-phosphate (S1P) [34]. SphK1 is localized predominantly in the cytosol, whereas SphK2 is localized mainly in the nuclear membrane and cytoplasm [35,36]. S1P functions in both extracellularly and intracellularly [37]. Extracellularly, S1P engages with five G protein-coupled receptors, S1PR1-5, to elicit prosurvival signaling [38,39,40]. Ceramide is involved in cell growth arrest and apoptosis, whereas S1P has the opposing function of promoting cell growth and cell survival, suggesting that the balance of ceramide and S1P is essential for cell fate [41]. S1P is metabolized by S1P phosphatase to sphingosine or by S1P lyase 1 to yield ethnolamine-1-phosphate and C16 fatty aldehyde. In the Golgi apparatus, ceramide is converted to sphingomyelin by sphingomyelin synthase (SMS) or to glucosylceramide by glucosylceramide synthase (GluCS), and then, to complex glycosylsphingolipids [42,43]. Phosphorylation of ceramide by ceramide kinase (CerK) produces ceramide-1-phosphate (C1P) [44]. C1P activates cytosolic phospholipase 2 (cPLA2), which recruits it to the Golgi apparatus and the cell membrane to cleave arachidonic acid and produce prostaglandin [45,46,47]. C1P also stimulates reactive oxygen species (ROS) formation in primary marrow-derived macrophages and ROS are required for the mitogenic effect of C1P [48].

## 3. Functions of Ceramide 

Ceramide is composed of a sphingosine base and amide-linked acyl chains varying in length from C14 to C26. CerS has six isoforms, CerS1-6, and the ceramides synthesized by these isoforms have different fatty acyl chains and distinct biological properties [21,49,50]. CerS1 preferentially generates ceramide with 18-carbon fatty acids, C18-ceramide, whereas CerS5 or CerS6 primarily generate ceramide 16-carbon fatty acids, C16-ceramide. A proapoptotic role of CerS1-generated C18-ceramide and prosurvival role of CerS6-generated C16-ceramide were suggested. Indeed, in HNSCC cells, CerS6-generated C16-ceramide protects cells from ER stress-mediated apoptosis and C18-ceramide is selectively down-regulated [49,51,52]. Karahatay et al. [52] measured ceramide levels in tumor tissues of 45 patients with HNSCC. The levels of C16- and C24-ceramides were significantly increased in the majority of tumor tissues compared with their normal tissues, whereas the levels of only C18-ceramide were significantly decreased in tumors. The decrease in C18-ceramide levels was also associated with the higher incidence of lymph node metastasis. Treatment with gemcitabine and doxorubicin, known inducers of ceramide generation, supra-additively inhibited the growth of human HNSCC cells. The treatment resulted in the elevation of mRNA and protein levels of the human homologue of yeast longevity assurance gene 1 (LASS1), which was consistent with an increase in the endogenous CerS activity for generation of C18-ceramide [53].

When the production and accumulation of ceramide increase in response to cellular stress, cell death is induced through numerous mechanisms such as apoptosis, necroptosis, autophagy, and ER stress [54]. These cell death pathways depend on cell type, the subcellular location of ceramide, and downstream targets of ceramide. Many types of cellular stress stimulate acid SMase (aSMase) to produce ceramide from sphingomyelin, leading to the formation of ceramide-enriched membrane platforms. In such platforms, nicotinamide adenine dinucleotide phosphate (NADPH) oxidase subunits and other redox molecules are aggregated, leading to signal transduction by the increase in ROS [55,56,57,58] (Figure 2). In neuronal cells, TNF-α exposure dramatically increased neutral SMase activity, thus, generating ceramide essential for subsequent NADPH oxidase activation and oxidative stress [59,60]. If the level of ROS increases further, it will damage DNA, proteins, and glycols. Mitochondria were demonstrated as a primary target of ceramide, leading to the generation of ROS by interacting with complex III of the electron transport chain [61,62].

Ceramide and activated Bax directly interact to induce outer membrane permeabilization in isolated mitochondria and induce Bax-dependent apoptosis [63]. Protein phosphatases, together with protein kinases, control the reversible phosphorylation of proteins, playing a central role in cellular signal transduction [64,65]. One of the well-described downstream targets of ceramide is the protein phosphatase 2A (PP2A) that regulates vital cellular processes, including cell cycle, growth, metabolism, and apoptosis [65]. SET is a nuclear protein and known inhibitor of PP2A activity. A unique mechanism for ceramide-mediated PP2A activation is the direct binding of ceramide to SET, relieving PP2A from SET, increasing PP2A activity, and leading to tumor suppression. C2-ceramide induces the PP2A-dependent dephosphorylation of Bcl-2 to inhibit its antiapoptotic activity and promote its binding to p53 for the induction of apoptosis [66,67,68].

Protein kinase C (PKC)ζ is an additional direct effector of ceramide implicated in ceramide-mediated growth arrest and apoptosis [69,70,71]. There are nine mammalian isoforms of PKC, which consist of the classical PKC (PKCα, β, γ), the novel PKC (PKCδ, ε, θ, η), and the atypical PKC (PKCζ, ι) [72,73,74]. The identification of PKC as the major receptor for tumor promoting phorbol esters marked the concept that PKC functions as an oncoprotein [69,75]. However, although phorbol esters result in acute activation of PKC, this is followed by the chronic loss or down-regulation of PKC. The repetitive application of phorbol esters is expected to cause a loss of PKC, promoting tumor formation. PKCζ inhibits Akt phosphorylation and activates the stress-induced protein kinase JNK [70,71,76]. In PKC mutations of cancers, loss-of-function with no gain-of-function was often identified [77].

Ceramide has been implicated in the induction of autophagy, which has dual functions in cell survival and cell death. Ceramide is a well-established suppressor of Akt [78], which inactivates downstream mammalian target of rapamycin (mTOR) signaling and leads to the induction of autophagy [79,80]. Ceramide also activates JNK and the JNK-mediated activation of the transcription factor c-Jun increases Beclin1 expression [81]. C-Jun positively regulates the transcription of LC3 to increase the autophagic process in response to ceramide [82]. In HNSCC and acute myeloid leukemia cells, stress-induced mitophagy is dependent on the localization of de novo-generated C18-ceramide at the outer mitochondrial membrane, which directly binds LC3 protein. Ceramide-LC3 binding results in the recruitment of autophagosomes for the execution of mitophagy [83,84].

Ceramide-mediated autophagy is also attributed to the induction of ER stress. The ER is the largest organelle in the cell and performs a variety of functions, including the synthesis of lipids, regulation of intracellular calcium, and synthesis and maturation of secreted and membrane-bound proteins, generating correctly folded proteins [85,86,87]. An increased need for protein-folding components is defined as ER proteotoxic stress. This stress induces the unfold protein response (UPR). The mammalian UPR consists of three parallel signaling pathways, initiated by the ER transmembrane sensors inositol requiring enzyme 1 (IRE1), PKR-like endoplasmic reticulum kinase (PERK), and activating transcription factor-6 (ATF6) [88,89,90,91]. Accumulation of long-chain ceramides, such as C14- and C16-ceramides, induces ER stress-dependent autophagy and functions in cell survival [92]. NADPH oxidase 4, an ER resident capable of producing ROS, acts as a signaling intermediate to transduce ER stress to the UPR. Chronic ER stress causes a secondary increase in ROS, generally resulting in cell death [93,94,95].

## 4. Radiation Therapy and Ceramide Generation

Ionizing radiation with X-rays stimulates cells to generate ceramide, an established second messenger in apoptotic signaling pathways. In irradiated cells, ceramide is mainly generated via the hydrolysis of sphingomyelin by SMase or by CerS-mediated synthesis [96,97,98]. Unlike the fast generation of ceramide at the plasma membrane via SMase, engagement of CerS and ceramide neogenesis is delayed, and increases between 8 and 24 h after irradiation [99,100]. In HeLa cells, dose-dependent ceramide generation was induced 28 h after irradiation, with a 2.3-fold increase by 10 Gy but 1.25-fold by 5 Gy [99]. Apoptosis of vascular endothelial cells in tumors and abnormal microvascular function lead to cell death of tumor cells. High doses of radiation can induce the translocation of endothelial cell aSMase into glycosphingolipid-enriched plasma membrane rafts, where it hydrolyzes sphingomyelin to generate ceramide. In contrast, the endothelial cell damage induced by the low-dose exposure in fractionated radiotherapy does not efficiently increase tumor cell death, as the death signaling pathway in endothelium is repressed by concomitant activation of tumor cell hypoxia inducible factor 1 (HIF-1). The post-radiation translation of HIF results in the up-regulation of vascular endothelial growth factor (VEGF) and other proangiogenic factors [101]. Single-dose radiotherapy (SDRT), a disruptive technique that ablates more than 90% of human cancers, operates a distinct dual-target mechanism, linking aSMase-mediated microvascular perfusion defects to DNA unrepair in tumor cells to confer tumor cell lethality [102].

CDases are overexpressed in several forms of cancer, including head and neck cancer, prostate cancer, and melanoma. Overexpression of acid CDase (aCDase) was observed in 70% of HNSCC patients, with an increased incidence of overexpression in higher grade tumors [103]. In prostate cancer, 5-Gy ionizing radiation induces increased sphingolipid expression between 24 and 72 h. Specifically, total ceramide content increased between 40% and 100%, whereas the products of ceramide metabolism, sphingosine and S1P, demonstrated a 200% to 270% increase, suggesting that irradiation elicited a marked shift of sphingolipid content toward the soluble products of ceramide metabolism [104]. The radiation-induced increase in aCDase activity confers resistance to Taxol and C6-ceramide [105,106]. 

## 5. Chemotherapy and Ceramide Generation

Chemotherapeutic agents can apply stress to cancer cells, and in response to this stress, ceramide levels are increased by both sphingomyelin hydrolysis and through de novo ceramide synthesis [21]. This increase, demonstrated by many chemotherapeutics, including daunorubicin (anthracycline), etoposide (topoisomerase II inhibitor), and gemcitabine (nucleoside analog), can be ascribed to the activation of CerS or SPT [107,108,109]. Vinblastine and paclitaxel were reported to increase ceramide levels and caspase activity in doxorubicin-resistant MCF-7-AdrR breast cancer cells [110]. Gemcitabine and doxorubicin combination reconstituted levels of C18-ceramide vis up-regulation of CerS1 expression, increasing CerS1 activity for C18-ceramide generation, leading to growth inhibition in head and neck xenografts in vivo [53]. In a phase II clinical trial, the combination therapy with gemcitabine and doxorubicin could represent an effective treatment for some patients with recurrent or metastatic HNSCC, and that serum C-18-ceramide elevation might be a serum biomarker of chemotherapy responses [111]. Up-regulation of GluCS levels prevents the accumulation of a ceramide pool, which reduces ceramide-induced apoptosis in response to certain cytotoxic drugs [18]. Doxorubicin treatment was reported to increase GluCS expression in invasive ductal breast cancer cells through the recruitment of transcription factor Sp1 to the GluCS promoter [112]. Cisplatin is widely used in many types of human solid neoplasms, including HNSCC, but the main limitation of its clinical usefulness is the high incidence of chemoresistance. Up-regulation of GluCS and P-glycoprotein (P-gp) leads to cisplatin resistance in head and neck cancer [113,114].

## 6. Inhibitors of Ceramide- and Sphingosine-Metabolizing Enzymes

### 6.1. CDase Inhibitor

A number of ceramide analogs, including LCL204, LCL385, and LCL521, were previously designed to target aCDase in the lysosomal compartment where aCDase primarily functions [115,116,117,118]. LCL204 alone was not toxic, but pretreatment of HNSCC cells with LCL204 significantly increased Fas-induced toxicity and apoptosis induction [103]. LCL385 was demonstrated to increase ceramide levels and sensitize PPC-1 prostate cancer cells to radiation, and significantly reduce tumor xenograft growth [119]. Photodynamic therapy (PDT) is a clinically established treatment modality for cancer. In PDT, a light-absorbing agent is activated by a highly focused laser to induce oxidative stress and destroy a cellular target [120]. When PDT-treated mouse SCCVII cells were used to vaccinate SCCVII tumor-bearing mice, adjuvant LCL521 treatment resulted in the marked retardation of tumor growth. This effect may have been due to the ability of LCL521 to restrict the activity of regulatory T cells (Tregs) and myeloid-derived suppressor cells (MDSCs) [121]. A novel CDase inhibitor, Ceranib-2, inhibited CDase activity, increased ceramide levels, reduced S1P levels, and inhibited cell proliferation and tumor growth [118]. Ceranib-2 and carboplatin exhibit synergism in combination for non-small cell lung cancer, where caspase 3, caspase 9, and Bax expression were increased, whereas Bcl-2 expression was reduced [122]. Treatment of acute myeloid leukemia cell lines with a novel aCDase inhibitor, SACLAC, effectively blocked aCDase activity and induced a decrease in S1P and an increase in total ceramide levels [123].

### 6.2. GluCS Inhibitor

D-threo-1-phenyl-2-palmitoylamino-3-morpholino-l-propanol (PPMP) and D-threo-1-phenyl-2-decanoylamino-3-morpholino-1-propanol (PDMP) were previously developed as GluCS inhibitors [124,125]. Exposure of MCF-7-AdrR breast cancer cells to PPMP reduced cellular ganglioside levels, restored sensitivity to vinblastine, increased vinblastine uptake three-fold, and reduced the expression of multidrug resistance 1 (MDR1) by 58% compared with untreated controls [126,127]. GluCS and P-gp overexpression is also associated with acquired cisplatin resistance in head and neck cancer. PPMP induced the accumulation of ceramide and increased cisplatin-induced cell death via P-gp down-regulation and restoration of p53-dependent apoptosis [113]. The antiestrogen tamoxifen, a first-generation P-gp inhibitor, blocked C6-ceramide glycosylation and magnified apoptotic responses. Specific high-affinity P-gp inhibitors, tariquidar and zosuquidar, synergistically enhanced C6-ceramide cytotoxicity in multidrug resistant leukemia cells [128]. A drug cocktail of GluCS inhibitor, CDase inhibitor, and imipramine that disturbs lipid turnover in biological membranes was reported to induce a marked increase in ceramide levels in radio-resistant HNSCC cells, corresponding with marked effects on radiation sensitivity [129]

### 6.3. SphK Inhibitor

Sphingosine kinases modulate the proliferation, apoptosis, and differentiation of keratinocytes through the regulation of ceramide and S1P. SphK1 expression has been shown to be elevated in HNSCC as compared with normal tissue, and positive SphK1 expression was associated with shorter survival time [130]. SphK1 is found to be the target of the microRNA, miR-124, which acts as a suppressor of HNSCC by directly inhibiting SphK1 activity and its downstream signaling [131]. These findings suggest that SphK1 inhibitors could be used to treat HNSCC Inhibitors for both SphK1 and SphK2 have been developed. They include N,N-dimethylsphingosine (DMS), SK1-I, SK-I-II, safingol, PF543, FTY720, and ABC294640 [132,133,134,135,136].

#### 6.3.1. Safingol

Safingol, the synthetic L-threo-stereoisomer of the naturally occurring (D-erythro-) dihydrosphingosine [137], was initially reported to be a PKC inhibitor, but was later found to inhibit SphK [138]. As other PKC inhibitors were not effective on cancer in clinical trials [74], the antitumor ability of safingol is not due to its anti-PKC activity. Safingol is a competitive inhibitor of SphK with a Ki of approximately 5 μM [139]. In MDA-MB-231 breast cancer cells and HT-29 colon cancer cells, 5–10 μM safingol reduced glucose uptake, activated AMP-activated protein kinase, and induced autophagy. In addition, Bcl-xL expression was decreased and Bax expression was increased, resulting in ROS-mediated necrotic cell death [140]. At the doses of 25–50 μM, safingol induced ROS-mediated apoptosis of human oral SCC cells [141]. During this process, Bcl-xL expression was decreased and mitochondrial endonuclease G was released into the cytoplasm, resulting in DNA fragmentation in the nucleus [142,143]. Autophagy was also induced, which promotes cell survival because autophagy inhibitors increase cell death by safingol [144]. Treatment of human colon cancer cells with 12 μM safingol increased the amounts of cell-associated safingol and N-acyl-safingol. In addition, small increases in endogenous sphingosine and dihydrosphingosine were noted, whereas no increases were observed in ceramide or dihydroceramide [145]. In response to ER stress, levels of dihydroceramide and dihydrosphingosine increased and, conversely, the addition of exogenous dihydroceramide or dihydrosphingosine activated on the ER stress sensor, ATF6, in HEK293 cells [146]. ATF6 can induce the objective genes, *Bip/Grp78, Grp94*, and C/EBP homology protein (CHOP) [147,148]. The transcription factor CHOP is induced late in ER stress, leading to increased expression of ER oxidase 1 and the generation of ROS within the ER [94]. In addition to the endogenous dihydrosphingosine increase by safingol treatment, safingol, the L-thero-dihydrosphingosine, may function as dihydrosphingosine intracellularly, leading to the induction of ER stress, ROS production, and cell death (Figure 3A). Safingol was the first SphK inhibitor to enter clinical trials as an anticancer agent [149,150]. In a phase I clinical trial, patients with advanced solid tumors were treated using safingol in combination with cisplatin [150]. Patients treated using a dose at or near the maximum tolerated dose achieved safingol levels of higher than 20 μM and maintained levels higher than 5 μM for 4 h. A dose-dependent reduction in S1P in plasma was observed. Reversible dose-dependent hepatic toxicity was also noted. The best response was stable disease in 6/37 (16%) for an average of 3.3 months (range 1.8–7.2 months). Safingol synergistically sensitized epigallocatechin-induced apoptotic cell death, and suppressed multiple myeloma cells by preventing protein tyrosine kinase phoshorylation and activation of death-associated protein kinase 1 (DAPK1) [151]. As high SphK1 correlates with resistance to cisplatin in gastric cancer, safingol was used synergistically with cisplatin to restore the efficacy of the chemotherapy in gastric cancer cells [152].

#### 6.3.2. PF543

PF543, a sphingosine-competitive cell-permeant inhibitor of SphK1, inhibits Sphk1 with a Ki of 3.6 nM. Its selectivity is more than 100-fold greater for SphK1 than for SphK2 [153]. However, PF543 had no inhibitory effects on the proliferation or survival of 1483 HNSCC cells, regardless of the marked decrease in the S1P/sphingosine ratio. Why PF543 did not exert antiproliferative effects was not clarified. The increased binding affinity of PF543 to SphK1 may result in a lack of specificity toward other enzymes, such as CerS, which effectively negates the effects of inhibiting SphK1 activity on cell growth and survival by preventing the formation of ceramide from sphingosine [154]. However, in 1483 cells treated with 3–10 μM PF543 for 16 h, PF543 did not inhibit CerS, increasing the amount of available sphingosine and synthesis of ceramides, including C18-ceramide and C24-ceramide. Long-chain ceramides, such as C18-ceramide, are antiproliferative, whereas very-long-chain ceramides, such as C24-ceramide, promote cell proliferation [155]. The variety of ceramide species generated by PF543 may determine its effects on cell viability. Alternatively, as an off-target effect, PF543 may inhibit mixed lineage kinase 1 (MLK1) at low concentrations [153]. The MLK family functions in JNK activation and TGF-β-induced cell death [156,157]. The cytotoxicity of PF543 by inhibiting SphK1 activity may be attenuated by its effects on the MLK family. On the other hand, at a concentration of 2.5 μM or higher, potent antiproliferative and cytotoxic effects of PF543 were demonstrated in human colorectal cancer cells, leading to necroptosis. In animal studies, intravenous injection of PF543 significantly suppressed HCT-166 xenograft growth while markedly improving mouse survival [158]. Treatment of oral SCC cells with PF543 at a concentration of 25 μM reduced cell viability, and induced apoptosis, necrosis, and autophagy, although autophagy promoted cell survival [159]. At higher concentrations, the inhibitory effects of PF543 on cell proliferation and cell survival may be ascribed to its off-target effects on other cellular enzymes, including SphK2. Levels of SphK1 in triple-negative breast cancer patients were significantly higher than those in patients with other breast tumors. PF543 sensitized such breast cancer cells to 5-FU and doxorubicin [160,161]. A PF543 derivative, Compound-2, has inhibitory activity on SphK1 in a manner similar to that of PF543, but it exerted antitumor activity on HT29 and HCT116 colorectal cancer cells at low concentrations [162].

#### 6.3.3. FTY720 (Fingolimod)

FTY720 is a sphingosine analog that functions via intracellular and extracellular pathways [163,164,165]. FTY720 is phosphorylated by SphK2 to become p-FTY720, and down-regulates S1PRs and inhibits T lymphocyte leakage in lymphoid tissues, marking it an immunosuppressant for recurrent multiple sclerosis [166,167]. S1PR inhibition by p-FTY720 sensitized drug-resistant colorectal cancer cells and tumors to cetuximab [25]. At higher concentrations (5–10 μM), FTY720 was demonstrated to be a potent apoptosis inducer in prostate, liver, and bladder cancer cells [165,168,169]. The antitumor effects of FTY720 were suggested to be due to its ability to stimulate ROS production, which culminated in PKCδ activation and subsequent caspase-3-dependent apoptosis in hepatocellular carcinoma cells [170] (Figure 3B). FTY720 also activates PP2A, which dephosphorylates proteins, with the most well-defined targets being Akt, Erk, c-Myc, and β-catenin, all of which are known to play a role in tumorigenesis [171,172]. In oral SCC, FTY720 inhibited Akt/NF-κB signaling, facilitated the proteasomal degradation of the antiapoptotic protein Mcl-1, and increased ROS generation, leading to apoptosis. When FTY720-induced autophagy was blocked by the autophagy inhibitor bafilomycin A1, FTY720-induced cell death was suppressed. This suggests that FTY720 induces autophagic cell death in oral SCC cells [173]. FTY720 acts synergistically with cisplatin to induce cell death [174]. In contrast to FTY720, OSU-2S was not phosphorylated by SphK2 in vitro and did not cause S1PR internalization in hepatocellular carcinoma cells or T lymphocyte homing in immunocompetent mice. OSU-2S suppressed the proliferation of hepatocellular carcinoma via ROS/PKCδ/caspase signaling and exhibited high in vivo potency in suppressing xenograft tumor growth without overt toxicity [175]. Furthermore, another FTY720 analog, Compound 7, which activates PP2A with antitumor activity was previously developed based on a PP2A docking study [176].

#### 6.3.4. ABC294640 (Opaganib)

ABC294640, 3-(4-chlorophenyl)-adamantane-1-carboxylic acid (pyridin-4-ylmethyl)amide, is a small-molecule SphK2-selective inhibitor that is orally available, has favorable pharmacological properties, and can reach therapeutic levels in mouse plasma and tumors without overt toxicity [136,177,178]. ABC294640 reduces S1P levels and increases ceramide in tumor cells, suppresses signaling through Akt, Erk, and NF-κB, increases proteasomal degradation of c-Myc, and promotes autophagy and/or apoptosis in a variety of cancer cells. In addition, ABC294640 induces proteasomal degradation of SphK1 and DES [179] (Figure 3C). As dihydroceramide was reported to be involved in ER stress [146], accumulation of dihydroceramide by DES inhibition may induce ER stress and ROS production, and function in the antitumor effects of ABC294640 [136,180]. In a phase I clinical study, ABC294640 was orally administered continuously in cycles of 28 days to 21 patients with advanced solid cancer [181]. The most common drug-related toxicities were nausea, vomiting, and fatigue. Among the patients evaluated, one (6%) had a partial response, six (38%) had stable disease, and nine (56%) had progressive disease as their best response. ABC294640 increased the transcription of pro-apoptotic Noxa and degradation of prosurvival Bcl-2 family molecule Mcl-1 in human cholangiocarcinoma cells. ABC294640 synergized with Bcl-2/Bcl-xL inhibitors ABT-263 and Obatoclax to induce cholangiocarcinoma cell death [182,183].

### 6.4. CerK Inhibitor

The CerK/C1P pathway has been implicated in survival signaling for cancer progression [184]. Exogenous C1P at low concentrations was reported to increase the survival and proliferation of NIH3T3 fibroblasts and A549 lung cancer cells, but at high concentrations, it reduced survival and induced apoptosis in correlation with the conversion of C1P to proapoptotic ceramide [185]. In MCF-7 breast cancer and NCI-H358 lung cancer cells, the CerK inhibitor NVP-231 caused a concentration-dependent decrease in cell viability and induced apoptosis. Cell cycle analysis revealed that NVP-231 reduced the number of cells in S phase and induced M phase arrest [186]. CerK is up-regulated in metastatic breast cancer cells, and plays a role in migration and invasion. The increased migration of CerK overexpressing cells was mitigated by NVP-231 via inhibition of the phosphoinositide 3-kinase (PI3K)/Akt pathway and Rho kinase, but not by inhibition of the classical Erk pathway [187].

## 7. Anti-S1P Antibody

Antibodies against S1P have been generated to block the S1P-mediated growth signal [188,189]. Sonepcizumab is a fully humanized monoclonal antibody directed at S1P. In a phase I study for treatment-resistant solid tumors, no dose-limiting adverse events were observed [190]. A phase II study was performed in patients with metastatic renal carcinoma with a history of prior VEGF-directed therapy [191]. Although the study did not achieve its primary endpoint based on two-month progression-free survival, a median overall survival of 21.7 months was observed. Four (10%) of 40 patients demonstrated a partial response, with a median duration of response of 5.9 months. No grade 3/4 treatment-related adverse events were observed. The most frequent grade 1/2 treatment-related adverse events were fatigue (30%), weight gain (18%), constipation (15%), and nausea (15%).

## 8. Synthetic Ceramides

Short-chain ceramides, unlike long-chain ceramides, can induce cell death, which is useful for therapeutic applications in cancer [192,193]. However, decreased solubility and bioavailability of ceramides analogues present problems for their delivery as chemotherapeutic agents in vivo. To overcome these problems, varied-chain pyridinium ceramides, such as C6-, C16-, or C18-pyridinium ceramides, have been synthesized with increased solubility and cell membrane permeability [194,195]. When PDT was combined with C6-pyridinum ceramide (LCL29), apoptosis via mitochondrial ceramide accumulation was increased [193]. A cationic water-soluble L-thero-C6-pyridinium-ceramide bromide (Ws-CER) inhibited the growth of several HNSCC cell lines, independent of their p53 status. Consistent with its targeting ability to negatively charged intracellular compartments, Ws-CER accumulated mainly in mitochondria- and nuclei-enriched fractions from treated cancer cells [192]. The combination of Ws-CER with gemcitabine further suppressed HNSCC in mice. Liposome-mediated delivery of C6-ceramide was reported to be an effective drug delivery method. Nanoliposomal C6-ceramide inhibited the in vitro and in vivo growth of breast cancer, pancreatic cancer, chronic lymphocytic leukemia, hepatocellular carcinoma, ovarian cancer, and melanoma [196,197,198,199,200,201]. Nanoliposomal C6-ceramide induced the phosphorylation of phosphoinositide 3-kinase (PI3K) and PKCζ, and dephosphorylation of PKCα. Concomitantly, activated PKCζ caused the dephosphorylation of paxillin, resulting in stress fiber depolymerization and focal adhesion disassembly in the metastatic tumor cells [202]. The combined use of nanoliposomal C6-ceramide with sub-therapeutic concentrations of gemcitabine exhibited cell toxicity in PANC-1, a gemcitabine-resistant human pancreatic cancer cell line [203].

## 9. Future Perspectives

A characteristic of treatment for HNSCC is the application of photon radiotherapy with gamma rays and X-rays. Although it is possible to irradiate a high dose of 6–13 Gy equivalent in particle radiotherapy using proton or carbon beams [204,205], each irradiation dose in fractionated X-ray radiotherapy for head and neck cancer is usually 2 Gy [206]. Previous studies reported that ceramide production in tumors and tumor vessels was low at an irradiation dose of 2 Gy [99,207]. Therefore, it is necessary to further investigate the interaction between the X-ray irradiation dose and ceramide production. PDT was reported to produce ceramide in the treatment of oral SCC, and the combined use of synthetic ceramide further increases the ceramide level and antitumor effects [193]. However, its effects may be limited to the superficial layer of advanced tumors. Microbubbles are often composed of microscopic lipid or protein shells encapsulating gaseous content, such as octafluoropropane, and have previously been used for imaging, gene delivery, tumor ablation, and as disrupting agents [208]. Recent studies using preclinical models of human tumors demonstrated that ultrasound-stimulated microbubbles (USMB) can mechanically perturb cell membranes, resulting in additive increases in ceramide-based radiation effects [209,210]. This method was confirmed to accumulate ceramide in endothelial, leukemia, breast cancer, prostate cancer, and fibrosarcoma cells [211,212], and has the advantage of producing ceramide at deeper tumors. Indeed, when sarcoma-bearing mice were treated with varying radiation doses with prior USMB exposure, acute vascular effects were induced, resulting in extensive tumor cell death. This was caused by increased ceramide production and can be elicited at low radiation doses (<8 Gy) by prior USMB exposure [213]. For distant metastasis unable to be treated by radiotherapy, chemotherapy with cisplatin or paclitaxel may be combined with USMB to increase the ceramide production at the site of ultrasound exposure. On the other hand, Alphonse et al. [129] demonstrated that a drug cocktail of GluCS inhibitor, CDase inhibitor, and imipramine increased ceramide levels in HNSCC cells, with marked effects on radiation sensitivity. This suggests that inhibitors of GluCS, CDase, SphK, or CerK can be used simultaneously to improve therapeutic outcomes. 

## 10. Application of Sphingolipid Target Therapy to HNSCC

The aCDase is overexpressed in 70% of HNSCC compared to normal controls (Section 4). Therefore, the aCDase inhibitor, LCL204, which can sensitize HNSCC cells to Fas-induced apoptosis in vitro and in vivo, could be a therapeutic agent for HNSCC (Section 6.1). SphK1 expression was also shown to be elevated in head and neck SCC compared to normal tissue, and positive SphK1 expression was associated with shorter survival time. SphK1 was found to be a target of the microRNA, miR-124, which acts as a suppressor in HNSCC by directly inhibiting SphK1 activity and downstream signaling (Section 6.3). These findings suggest that SphK1 inhibitors could be used to treat HNSCC. Indeed, in a phase I study, treatment of solid tumors containing HNSCC with safingol and cisplatin effectively down-regulated S1P (Section 6.3.1). Synthetic ceramides, such as C6-pyridium ceramide, enhanced the ability of PDT to accumulate ceramide in mitochondria, followed by cytochrome c release and caspase 3 activation. Therefore, the utility of C6-pyridium ceramide in the treatment of HNSCC with PDT was suggested (Section 8). As a clinical study, a phase II clinical trial revealed that combination therapy with gemcitabine and doxorubicin was an effective treatment for some patients with recurrent or metastatic HNSCC, and that serum C-18-ceramide elevation can serve as a serum biomarker to determine responsiveness to chemotherapy (Section 5). In another clinical trial, patients with recurrent HNSCC were treated with combination therapy of in vitro-expanded natural killer T (NKT) cells and α-galactosylceramide (αGalCer)-pulsed antigen-presenting cells. The intra-arterial infusion of Vα24NKT cells and the submucosal injection of αGalCer-pulsed APC induced significant antitumor immunity with beneficial clinical effects in the management of advanced HNSCC [214]. Thus, although clinical trials on sphingolipids in HNSCC have been limited, GluCS inhibitors, SphKs inhibitors (PF543, FTY720, and ABC294640), CerK inhibitors, anti-S1P antibodies, and synthetic ceramides (Section 6, Section 7 and Section 8) against HNSCC should be validated in clinical studies. Radiotherapy is a major treatment modality for HNSCC, with radiation increasing ceramide levels through hydrolysis of sphingolipid by SMase and CerS-mediated synthesis (Section 4). Because these reagents can be used to sensitize SCC to radiotherapy/ radiochemotherapy, HNSCC is in an advantageous situation compared to tumors that cannot be treated with radiation. 

## 11. Conclusions

Although a number of inhibitors for ceramide- and sphingosine-metabolizing enzymes have been developed, published clinical trials have been limited to several inhibitors, such as safingol and ABC294640, and the anti-S1P antibody sonepcizumab. The major role of these inhibitors may be to increase the sensitivity of HNSCC, including oral SCC, to radiotherapy and chemotherapy through the increase in ceramide levels in the tumor. When the X-ray radiation dose in fractionated radiotherapy is low, an additional method, such as prior USMB exposure, is helpful to increase ceramide generation. Based on studies using the SphK2-selective inhibitor ABC294640, it can inhibit DES and SphK2. This suggests the importance of off-target effects of the inhibitor in its antitumor activity. In order to improve the therapeutic effects of radiotherapy and chemotherapy for HNSCC, a cocktail of multiple inhibitors for ceramide- and sphingosine-metabolizing enzymes may be also useful.

## Figures and Tables

**Figure 1 cancers-12-02062-f001:**
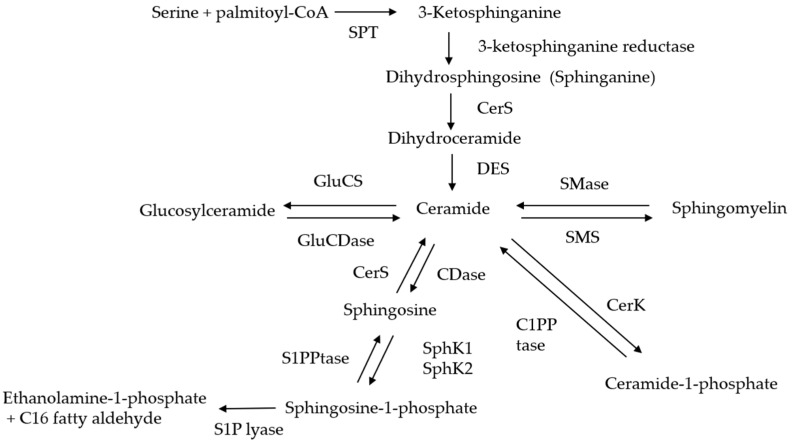
Ceramide-generating and -metabolizing pathways. Sphingomyelin present in the cell membrane is hydrolyzed by sphingomyelinase (SMase) to produce ceramide. Serine and palmitoyl-CoA were condensed by serine palmitoyl transferase (SPT) to form 3-ketosphinganine, which then becomes dihydroceramide by ceramide synthase (CerS). Dihydroceramide desaturase (DES) functions to form ceramide. Ceramide is metabolized by the action of ceramidase (CDase) and converted into sphingosine, which is phosphorylated by sphingosine kinase 1 (SphK1) and SphK2 to become sphingosine-1-phosphate (S1P). Glucosylceramide synthase (GluCS) glucosylates ceramide to yield glucosylceramide. Ceramide is also phosphorylated by ceramide kinase (CerK) and converted into ceramide-1-phosphate (C1P). SMS, sphingomyelin synthase; S1PPtase, sphingosine-1-phosphate phosphatase; C1PPtase, ceramide-1-phosphate phosphatase.

**Figure 2 cancers-12-02062-f002:**
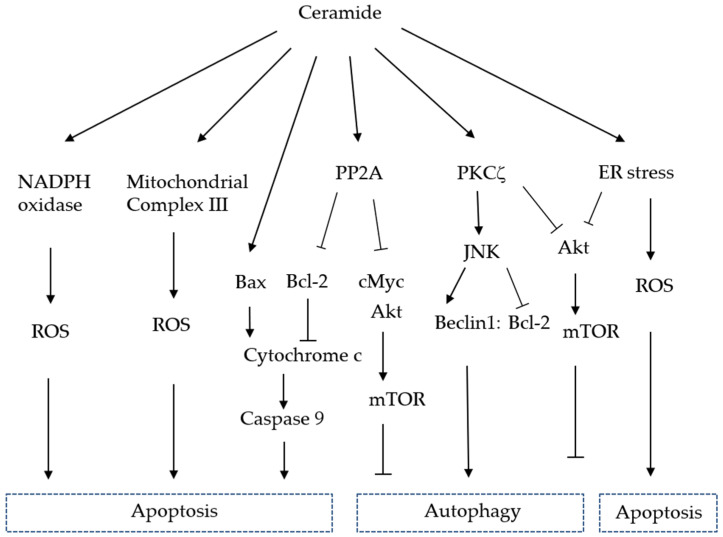
Outlines of ceramide-driven signaling pathways. Ceramide activates PP2A, which activates Bax and suppresses Bcl-2 to promote apoptosis. Ceramide suppresses Akt, inactivates downstream mTOR, and induces autophagy. Ceramide activates PKCζ, which inhibits Akt phosphorylation and activates JNK. Activated c-Jun increases Beclin1 expression. Ceramide also increases NADPH oxidase- and mitochondrial complex III-mediated ROS production. NADPH oxidase, nicotinamide adenine dinucleotide phosphate oxidase; PP2A, protein phosphatase 2A; ER stress, endoplasmic reticulum stress; ROS, reactive oxygen species; mTOR, mammalian target of rapamycin.

**Figure 3 cancers-12-02062-f003:**
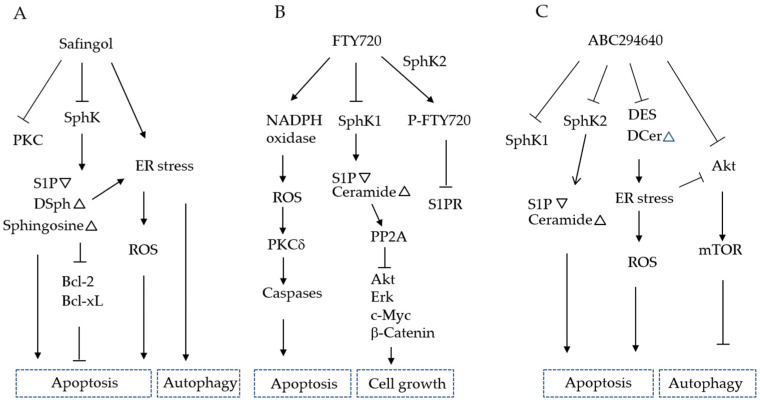
Proposed signaling pathways affected by SphK inhibitors. (**A**) Safingol inhibits SphK, and increases sphingosine and dihydrosphingosine (DSph), but not ceramide. These increased sphingolipids may induce ER stress and ROS production, leading to apoptosis and/or autophagy. (**B**) FTY720 is phosphorylated by SphK2, resulting in p-FTY720, which extracellularly inhibits S1PR signaling. Intracellularly, FTY720 inhibits SphK1, activates PP2A, and suppresses Akt, Erk, c-Myc, and β-catenin. FTY720 also activates PKCδ through an NADPH-mediated pathway. (**C**) ABC294640 inhibits SphK2, and induces proteasomal degradation of SphK1 and dihydroceramide desaturase (DES). Accumulated dihydroceramide (DCer) can induce ER stress, up mark(triangle), increase by the inhibitor, and down mark (inverted triangle), decrease by the SphK inhibitor.

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
