# Peer review of "Inhibitors of Ceramide- and Sphingosine-Metabolizing Enzymes as Sensitizers in Radiotherapy and Chemotherapy for Head and Neck Squamous Cell Carcinoma"

_cancers, 2020, doi:10.3390/cancers12082062_

Round 1

Reviewer 1 Report

This is a comprehensive review article to discuss ceramide biological function, metabolism and responses of cancer radiotherapy, chemotherapy and combination therapy. Specifically, the authors summarized the outcome of most popular inhibitors of ceramide-metabolizing enzymes, anti-S1P antibody and synthetic ceramides. The article is helpful for either general biomedical readers or cancer researchers. There are some suggestions as below:

  • In the introduction section, the authors summarized the molecular profiles of SCC from the TCGA data. Since the review paper is focused on ceramide, can any ceramide profiles be found from the TCGA data? It will be better to link to the article and next paragraph.
  • There seems to be a disconnect between paragraphs 2 and 3 in the introduction.
  • Based on the title of article, authors want to focus on oral squamous cell carcinoma. However, I didn’t see any strong distinction between head and neck cancer studies and the many studies the author discussed such as breast cancer, lung cancer and other cancers to support the content. Recommend generalizing the title name.

Author Response

Response to reviewer’s comments

[Reviewer 1]

This is a comprehensive review article to discuss ceramide biological function, metabolism and responses of cancer radiotherapy, chemotherapy and combination therapy. Specifically, the authors summarized the outcome of most popular inhibitors of ceramide-metabolizing enzymes, anti-S1P antibody and synthetic ceramides. The article is helpful for either general biomedical readers or cancer researchers.

Response: Thank you for your positive comments to our paper.

In the introduction section, the authors summarized the molecular profiles of SCC from the TCGA data. Since the review paper is focused on ceramide, can any ceramide profiles be found from the TCGA data?

Response: As indicated by the reviewer, it is important to know the profile of expression levels of ceramide-metabolizing enzymes using TCGA data, but we cannot find such articles on head and neck SCC in the literature.

It will be better to link to the article and next paragraph. There seems to be a disconnect between paragraphs 2 and 3 in the introduction.

Response: We have added a sentence indicating that other approaches are required (Introduction, page 2, lines 56-57).

Based on the title of article, authors want to focus on oral squamous cell carcinoma. However, I didn’t see any strong distinction between head and neck cancer studies and the many studies the author discussed such as breast cancer, lung cancer and other cancers to support the content. Recommend generalizing the title name.

Response: Following the suggestion, we have changed “oral squamous cell carcinoma” to “head and neck squamous cell carcinoma” in the title.

Reviewer 2 Report

The review article by Yura and colleagues aims to discuss how inhibitors of ceramide-metabolizing enzymes (although sphingolipid metabolism would be more accurate) might be used to sensitize tumours to the therapeutic effects of radiotherapy and chemotherapy.  The title suggests the review will focus on oral squamous cell carcinoma (OSCC), but this emphasis isn’t always obvious and clear. There have been some recent comprehensive reviews discussing similar pathways (eg Ogretmen, Nat Rev Cancer 2018; Kroll et al Frontiers in Oncology 2020), so to conentrate on a particular cancer such as OSCC potentially adds some novelty to the article. However, it isn’t clear whether the amount of data available for OSCC is sufficient to focus entirely on this disease, as the relevant data are intertwined with information from other tumour types throughout the manuscript, leaving the reader unsure of the potential of these inhibitors as therapeutic agents for OSCC. If there are sufficient data available related specifically to OSCC, the review would need to be refocused and reorganized to be suitable for publication

Specific comments

The title indicates that the review will focus on ceramide-metabolizing enzymes. This is slightly misleading as considerable attention is paid to SPHK inhibitors also, which phosphorylate sphingosine to produce S1P. Whilst the production of S1P will affect ceramide levels, SPHKs doesn’t metabolize ceramide directly. Sphinolipid metabolism might be a more accurate title.

The description of sphingolipid metabolism and the sphingolipid rheostat isn’t described in sufficient detail in the Introduction. This is needed to provide a bit more context. In a similar way, the figures as linear diagrams are not very contextual and could be made more informative and interesting.

The information relating specifically to OSCC is described in amongst data from other cancer types and is spread throught the various sections. As a result it gets lost in the “noise.” Further, it isn’t always clear how the OSCC data relates to radio and chemotherapy (as indicated in the title). For example, in Section 3 “functions of ceramide” discussion of the role of C16, C18 and C24 ceramide is the only data relating to OSCC in this section and some of this relates to metastasis. Similarly, in some other sections there is little or no mention of OSCC. The authors should consider an entirely separate section on OSCC and perhaps describe what is known about the sphingolipid pathway in OSCC, such as ceramide levels, SPHK1 expression etc and how these might be targeted with inhibitors.

The authors need to clarify when they are discussing HNSCC and when the data relate to OSCC specifically. There is always some confusion in the literature in this regard, however it is now known that HPV +ve oropharyngeal cancer (a subset of HNSCC) is essentially a distinct disease.

Some of the cited references appear to be wrong. For example, refs 34, 37 and 38 are supposed to relate to S1P receptors, but only reference 37 seems relevant. Other examples include refs 95 and 177, which appear to be not correct. The authors should carefully check all references to ensure accuracy and relevance.

The future perspectives and conclusions should be more focused towards OSCC.

Author Response

Response to reviewer’s comments

[Reviewer 2]

The review article by Yura and colleagues aims to discuss how inhibitors of ceramide-metabolizing enzymes (although sphingolipid metabolism would be more accurate) might be used to sensitize tumours to the therapeutic effects of radiotherapy and chemotherapy.  The title suggests the review will focus on oral squamous cell carcinoma (OSCC), but this emphasis isn’t always obvious and clear. There have been some recent comprehensive reviews discussing similar pathways (eg Ogretmen, Nat Rev Cancer 2018; Kroll et al Frontiers in Oncology 2020), so to conentrate on a particular cancer such as OSCC potentially adds some novelty to the article. However, it isn’t clear whether the amount of data available for OSCC is sufficient to focus entirely on this disease, as the relevant data are intertwined with information from other tumour types throughout the manuscript, leaving the reader unsure of the potential of these inhibitors as therapeutic agents for OSCC. If there are sufficient data available related specifically to OSCC, the review would need to be refocused and reorganized to be suitable for publication

Response: Thank you for helpful comments to the title. Following the suggestion, we have changed “oral squamous cell carcinoma” to “head and neck squamous cell carcinoma” in the title. The review of Ogretmen, Nat Rev Cancer 2018 had been cited in original manuscript (ref. 20). This time, the recent article of Kroll et al. has been added (ref. 29).

The title indicates that the review will focus on ceramide-metabolizing enzymes. This is slightly misleading as considerable attention is paid to SPHK inhibitors also, which phosphorylate sphingosine to produce S1P. Whilst the production of S1P will affect ceramide levels, SPHKs doesn’t metabolize ceramide directly. Sphinolipid metabolism might be a more accurate title.

Response: The word “splingolipid” covers so many molecules. We want to focus ceramide that plays a central role in cancer therapy. However, as indicated by the reviewer, sphingosine kinases are not included in ceramide-metabolizing enzyme. In the revised version, we have changed “ceramide-metabolizing enzymes” to “ceramide- and sphingosine-metabolizing enzymes” in the title.

The description of sphingolipid metabolism and the sphingolipid rheostat isn’t described in sufficient detail in the Introduction. This is needed to provide a bit more context. In a similar way, the figures as linear diagrams are not very contextual and could be made more informative and interesting.

Response: We have stated the balance of ceramide and S1P (Section2, line 89-91). As indicated by the reviewer, the figure legends are not enough. We have explained the figures with several sentences in revised version.

The information relating specifically to OSCC is described in amongst data from other cancer types and is spread throught the various sections. As a result it gets lost in the “noise.” Further, it isn’t always clear how the OSCC data relates to radio and chemotherapy (as indicated in the title). For example, in Section 3 “functions of ceramide” discussion of the role of C16, C18 and C24 ceramide is the only data relating to OSCC in this section and some of this relates to metastasis. Similarly, in some other sections there is little or no mention of OSCC. The authors should consider an entirely separate section on OSCC and perhaps describe what is known about the sphingolipid pathway in OSCC, such as ceramide levels, SPHK1 expression etc and how these might be targeted with inhibitors.

Response: We have changed the title to cover head and neck squamous cell carcinoma.

The authors need to clarify when they are discussing HNSCC and when the data relate to OSCC specifically. There is always some confusion in the literature in this regard, however it is now known that HPV +ve oropharyngeal cancer (a subset of HNSCC) is essentially a distinct disease.

Response: As indicated by the reviewer, characteristics of head and neck SCC are different depending on the status of HPV infection. However, we cannot obtain enough information about sphingolipid metabolism in HPV-positive and HPV-negative head and neck SCC.

Some of the cited references appear to be wrong. For example, refs 34, 37 and 38 are supposed to relate to S1P receptors, but only reference 37 seems relevant. Other examples include refs 95 and 177, which appear to be not correct. The authors should carefully check all references to ensure accuracy and relevance.

Response: Thank you for useful suggestions, we have deleted ref. 34 and 35 and replaced by more adequate articles (revised ref 35, 36) showing the intracellular localization of SphK1 and SphK2. Ref 38 has been also replaced by papers relating S1PR (ref. 39, 40). The text has been changed following the content of ref. 95. In section 8, we kept ref 177 (191 after revision), but added two papers indicating the solubility and cell membrane permeability of synthetic ceramides (ref. 193, 194).     

The future perspectives and conclusions should be more focused towards OSCC.

Response: We have changed the title.

Reviewer 3 Report

The manuscript by Yura et al. is a review focused on inhibitors of ceramide metabolism for potential use alongside chemotherapy and radiotherapy of oral squamous cell carcinoma. This is a reasonably-comprehensive review that highlights the major and important points associated with an important and emerging topic of therapeutic interest. Ceramide is a bioactive sphingolipid and its metabolism have long been studied along with the potential utility of ceramide-directed therapies. However, advancement of these drugs into the clinic has driven widespread scientific and clinical interest that demands helpful reviews such as this one by Yura et al. The following are important points that should be addressed and updated within the manuscript.

1. The authors begin Section 3 with a discussion centered on ceramide species and the CerS enzymes that generate them (first paragraph, lines 100-112). There have been some notable differences in effects that certain studies have attributed to different ceramide species. The authors do a modest job of discussing this. However, they have missed discussing the key study listed below which is focused on head and neck squamous cell carcinoma. This might be because the authors have cited other reviews, but the key and original research needs to be cited (as mentioned below in comment #11).

Senkal CE, Ponnusamy S, Rossi MJ, Bialewski J, Sinha D, Jiang JC, Jazwinski SM, Hannun YA, Ogretmen B.Role of Human Longevity Assurance Gene 1 and C18-ceramide in Chemotherapy-Induced Cell Death in Human Head and Neck Squamous Cell Carcinomas.Mol Cancer Ther. 2007 Feb;6(2):712-22. doi: 10.1158/1535-7163.MCT-06-0558

2. The authors discuss ceramide regulation of autophagy later in Section 3 (fourth paragraph, lines 144-149). This is a brief paragraph that lacks important information tying a key ceramide species (C18:0 ceramide), and the enzyme that generates it (CerS1), to autophagy of mitochondria (mitophagy). There are three important studies, two listed below, that the authors need to include in their review and correctly discuss. The authors do reference one important ceramide/mitophagy study (citation#76, line 149, Sentelle et al.), but incorrectly associate the study with the citation that follows (citation#77). Citation #77 focuses on a role for c-jun in ceramide-mediated autophagy, but Sentelle et al (citation#76) does not. Sentelle et al. should be discussed alongside the two other studies from the Ogretmen group, which are mentioned below. These collectively discuss ceramide regulation of mitophagy, including a recent mechanism involving p17/PERMIT-mediated movement of CerS1 to mitochondria.

Oleinik N, Kim J, Roth BM, Selvam SP, Gooz M, Johnson RH, Lemasters JJ, Ogretmen B. Mitochondrial Protein Import Is Regulated by p17/PERMIT to Mediate Lipid Metabolism and Cellular Stress. Sci Adv. 2019 Sep 11;5(9):eaax1978. doi: 10.1126/sciadv.aax1978

Dany M, Gencer S, Nganga R, Thomas RJ, Oleinik N, Baron KD, Szulc ZM, Ruvolo P, Kornblau S, Andreeff M, Ogretmen B. Targeting FLT3-ITD Signaling Mediates Ceramide-Dependent Mitophagy and Attenuates Drug Resistance in AML. Blood. 2016 Oct 13;128(15):1944-1958. doi: 10.1182/blood-2016-04-708750

3. The authors discuss ceramide regulation of the NADPH oxidase (NOX) in Section 3 (second paragraph, lines 113-123). However, this was incomplete and focused only on ceramide generated through the acid sphingomyelinase pathway in endothelial cells. The authors should also discuss the following study where the neutral sphingomyelinase has been shown to mediated NOX activity in neurons and neuroblastoma cells.

Barth BM, Gustafson SJ, Kuhn TB. Neutral Sphingomyelinase Activation Precedes NADPH Oxidase-Dependent Damage in Neurons Exposed to the Proinflammatory Cytokine Tumor Necrosis factor-α. J Neurosci Res. 2012 Jan;90(1):229-42. doi: 10.1002/jnr.22748

4. The authors should also discuss the two studies listed below, which demonstrated that the ceramide-metabolite C1P can regulate NOX. This would be a logical progression from their discussion of ceramide-mediation of NOX (see comment #3 above), as well as logical given the author’s later discussion of CerK inhibitors.

Barth BM, Gustafson SJ, Hankins JL, Kaiser JM, Haakenson JK, Kester M, Kuhn TB. Ceramide Kinase Regulates TNFα-stimulated NADPH Oxidase Activity and Eicosanoid Biosynthesis in Neuroblastoma Cells. Cell Signal. 2012 Jun;24(6):1126-33. doi: 10.1016/j.cellsig.2011.12.020

Arana L, Gangoiti P, Ouro A, Rivera IG, Ordoñez M, Trueba M, Lankalapalli RS, Bittman R, Gomez-Muñoz A. Generation of Reactive Oxygen Species (ROS) Is a Key Factor for Stimulation of Macrophage Proliferation by Ceramide 1-phosphate. Exp Cell Res. 2012 Feb 15;318(4):350-60. doi: 10.1016/j.yexcr.2011.11.013

5. Single-dose radiation therapy and the role of acid sphingomyelinase in its efficacy has been recently documented in a study that included a clinical trial. It is recommended that the authors include this study in their review in Section 4.

Bodo S, Campagne C, Thin TH, Higginson DS, Vargas HA, Hua G, Fuller JD, Ackerstaff E, Russell J, Zhang Z, Klingler S, Cho H, Kaag MG, Mazaheri Y, Rimner A, Manova-Todorova K, Epel B, Zatcky J, Cleary CR, Rao SS, Yamada Y, Zelefsky MJ, Halpern HJ, Koutcher JA, Cordon-Cardo C, Greco C, Haimovitz-Friedman A, Sala E, Powell SN, Kolesnick R, Fuks Z Single-dose Radiotherapy Disables Tumor Cell Homologous Recombination via ischemia/reperfusion Injury. .J Clin Invest. 2019 Feb 1;129(2):786-801. doi: 10.1172/JCI97631

6. FTY720 is discussed in Section 6.3.3 as a subsection for SphK inhibitors. This is somewhat misleading as it is not primary known as a SphK inhibitor. It is recommended that a separate section be created for S1PR regulators, which is what FTY720 is best recognized as and is its clinically-recognized/approved activity (for treating MS). A separate section for S1PR regulators would also allow the authors the opportunity to discuss other S1P regulators as mentioned below in comment #8. In addition, the authors should mention that FTY720 is now known as Fingolimod, which will help others more easily search for information about this drug

7. ABC294640 is discussed in Section 6.3.4. The authors should mention that ABC294640 is now known as Opaganib, which will help others more easily search for information about this drug. In addition, this section could be improved by mentioning that ABC294640 is currently in clinical trials for prostate cancer, cholangiocarcinoma (follow-up trials), as well as for COVID-19 (which presently is a very important topic).

8. The authors missed discussing some prominent inhibitors/regulators. These mostly are inhibitors or regulators of CDase (SACLAC), SphK (MP-A08, SK1-I), S1PR (CYM-5478), and GluCS/P-Gp (tamoxifen, zosuquidar). Some of the relevant literature that should be included is listed as follows.

Pearson JM, Tan SF, Sharma A, Annageldiyev C, Fox TE, Abad JL, Fabrias G, Desai D, Amin S, Wang HG, Cabot MC, Claxton DF, Kester M, Feith DJ, Loughran TP.Ceramide Analogue SACLAC Modulates Sphingolipid Levels and MCL-1 Splicing to Induce Apoptosis in Acute Myeloid Leukemia. Mol Cancer Res. 2020 Mar;18(3):352-363. doi: 10.1158/1541-7786.MCR-19-0619

Pitman MR, Powell JA, Coolen C, Moretti PA, Zebol JR, Pham DH, Finnie JW, Don AS, Ebert LM, Bonder CS, Gliddon BL, Pitson SM.A Selective ATP-competitive Sphingosine Kinase Inhibitor Demonstrates Anti-Cancer Properties. Oncotarget. 2015 Mar 30;6(9):7065-83. doi: 10.18632/oncotarget.3178

Powell JA, Lewis AC, Zhu W, Toubia J, Pitman MR, Wallington-Beddoe CT, Moretti PA, Iarossi D, Samaraweera SE, Cummings N, Ramshaw HS, Thomas D, Wei AH, Lopez AF, D'Andrea RJ, Lewis ID, Pitson SM.Targeting Sphingosine Kinase 1 Induces MCL1-dependent Cell Death in Acute Myeloid Leukemia. Blood. 2017 Feb 9;129(6):771-782. doi: 10.1182/blood-2016-06-720433

Pitman MR, Costabile M, Pitson SM.Recent Advances in the Development of Sphingosine Kinase Inhibitors. Cell Signal. 2016 Sep;28(9):1349-63. doi: 10.1016/j.cellsig.2016.06.007

Edmonds Y, Milstien S, Spiegel S.Development of Small-Molecule Inhibitors of sphingosine-1-phosphate Signaling. Pharmacol Ther. 2011 Dec;132(3):352-60. doi: 10.1016/j.pharmthera.2011.08.004

Wang W, Shanmugam MK, Xiang P, Yam TYA, Kumar V, Chew WS, Chang JK, Ali MZB, Reolo MJY, Peh YX, Karim SNBA, Tan AYY, Sanda T, Sethi G, Herr DR.Sphingosine 1-Phosphate Receptor 2 Induces Otoprotective Responses to Cisplatin Treatment. Cancers (Basel). 2020 Jan 15;12(1):211. doi: 10.3390/cancers12010211

Wang W, Xiang P, Chew WS, Torta F, Bandla A, Lopez V, Seow WL, Lam BWS, Chang JK, Wong P, Chayaburakul K, Ong WY, Wenk MR, Sundar R, Herr DR.Activation of Sphingosine 1-phosphate Receptor 2 Attenuates Chemotherapy-Induced Neuropathy. J Biol Chem. 2020 Jan 24;295(4):1143-1152. doi: 10.1074/jbc.RA119.011699

Morad SAF, Davis TS, MacDougall MR, Tan SF, Feith DJ, Desai DH, Amin SG, Kester M, Loughran TP Jr, Cabot MC.Role of P-glycoprotein Inhibitors in Ceramide-Based Therapeutics for Treatment of Cancer. Biochem Pharmacol. 2017 Apr 15;130:21-33. doi: 10.1016/j.bcp.2017.02.002

9. The authors do a reasonable job discussing ceramide-technologies in Section 8, which includes a modest but incomplete discussion about the ceramide nanoliposome. Given the focus and intent of their review, the authors should indicate that this technology is in clinical trials for cancer (Ceramide NanoLiposome: ClinicalTrials.gov Identifier: NCT02834611). The authors list some of the cancers that the ceramide nanoliposome has been shown to treat, but this is an incomplete list and the referencing is very minimal. More so, the authors only elaborate on one study (citation #184, Jiang et al.) where the ceramide nanoliposome was tested in combination with chemotherapy. There are many other studies evaluating the combination of the ceramide nanoliposome with chemotherapy. The authors should add the following studies, which are the most recent studies (acute myeloid leukemia, colorectal cancer, liver cancer immune responses, ovarian cancer) as well as a review of its preclinical development.

Barth BM, Wang W, Toran PT, Fox TE, Annageldiyev C, Ondrasik RM, Keasey NR, Brown TJ, Devine VG, Sullivan EC, Cote AL, Papakotsi V, Tan SF, Shanmugavelandy SS, Deering TG, Needle DB, Stern ST, Zhu J, Liao J, Viny AD, Feith DJ, Levine RL, Wang HG, Loughran TP Jr, Sharma A, Kester M, Claxton DF. Sphingolipid Metabolism Determines the Therapeutic Efficacy of Nanoliposomal Ceramide in Acute Myeloid Leukemia. Blood Adv. 2019 Sep 10;3(17):2598-2603. doi: 10.1182/bloodadvances.2018021295

Lu P, White-Gilbertson S, Nganga R, Kester M, Voelkel-Johnson C. Expression of the SNAI2 Transcriptional Repressor Is Regulated by C 16-ceramide. Cancer Biol Ther. 2019;20(6):922-930. doi: 10.1080/15384047.2019.1579962

Li G, Liu D, Kimchi ET, Kaifi JT, Qi X, Manjunath Y, Liu X, Deering T, Avella DM, Fox T, Rockey DC, Schell TD, Kester M, Staveley-O'Carroll KF. Nanoliposome C6-Ceramide Increases the Anti-tumor Immune Response and Slows Growth of Liver Tumors in Mice. Gastroenterology. 2018 Mar;154(4):1024-1036.e9. doi: 10.1053/j.gastro.2017.10.050

Zhang X, Kitatani K, Toyoshima M, Ishibashi M, Usui T, Minato J, Egiz M, Shigeta S, Fox T, Deering T,Kester M, Yaegashi N. Ceramide Nanoliposomes as a MLKL-Dependent, Necroptosis-Inducing, Chemotherapeutic Reagent in Ovarian Cancer. Mol Cancer Ther. 2018 Jan;17(1):50-59. doi: 10.1158/1535-7163.MCT-17-0173

Kester M, Bassler J, Fox TE, Carter CJ, Davidson JA, Parette MR. Preclinical Development of a C6-ceramide NanoLiposome, a Novel Sphingolipid Therapeutic. Biol Chem. 2015 Jun;396(6-7):737-47. doi: 10.1515/hsz-2015-0129

10. There are several minor errors as follows.

Page 2, line 84: spelling error “shingosine-1-phosphate”

Page 2, line 92: spelling error “sphigomyelin synthase”

Page 6, line 197: spelling error “CesS1”

Page 9, line 351: grammar error “has”

Figure 1: spelling error “3-ketosphinganie reductase”; “Dihydrosphindosine”

Figure 2: spelling error “Cytocrome”

Figure 3A: spelling error “sphigosine”

Special (Greek) characters are omitted throughout the manuscript.

Please double-check the manuscript for other possible spelling/grammar errors

11. The authors cite a substantial number of review articles rather than the primary literature. For reviews, it is usually best to cite the original research when possible. However, it is understandable that some reviews will ultimately be cited. This might be appropriate in some cases, especially when the reviews are prominent and/or summarizing a large body of work.

12. The focus of this review according to the title and abstract is on oral squamous cell carcinoma (SCC). However, much of the manuscript summarizes work that was not conducted on SCC. This is a minor concern that the authors might address in their abstract. It could be highlighted that much of the reviewed material, even if not specifically using SCC models, is directly-applicable to SCC as the fundamental target is ceramide metabolism.

Author Response

Response to reviewer’s comments

[Reviewer 3]

The manuscript by Yura et al. is a review focused on inhibitors of ceramide metabolism for potential use alongside chemotherapy and radiotherapy of oral squamous cell carcinoma. This is a reasonably-comprehensive review that highlights the major and important points associated with an important and emerging topic of therapeutic interest. Ceramide is a bioactive sphingolipid and its metabolism have long been studied along with the potential utility of ceramide-directed therapies. However, advancement of these drugs into the clinic has driven widespread scientific and clinical interest that demands helpful reviews such as this one by Yura et al.

Response: Thank you for your careful reading and positive comments to our paper.

  1. The authors begin Section 3 with a discussion centered on ceramide species and the CerS enzymes that generate them (first paragraph, lines 100-112). There have been some notable differences in effects that certain studies have attributed to different ceramide species. The authors do a modest job of discussing this. However, they have missed discussing the key study listed below which is focused on head and neck squamous cell carcinoma. This might be because the authors have cited other reviews, but the key and original research needs to be cited (as mentioned below in comment #11).

Senkal CE, Ponnusamy S, Rossi MJ, Bialewski J, Sinha D, Jiang JC, Jazwinski SM, Hannun YA, Ogretmen B.Role of Human Longevity Assurance Gene 1 and C18-ceramide in Chemotherapy-Induced Cell Death in Human Head and Neck Squamous Cell Carcinomas.Mol Cancer Ther. 2007 Feb;6(2):712-22. doi: 10.1158/1535-7163.MCT-06-0558

Response: Following the suggestion, we have added the content of the paper by Senkal et al. 2007 (ref. 53)(page 3, lines 114-118).

  1. The authors discuss ceramide regulation of autophagy later in Section 3 (fourth paragraph, lines 144-149). This is a brief paragraph that lacks important information tying a key ceramide species (C18:0 ceramide), and the enzyme that generates it (CerS1), to autophagy of mitochondria (mitophagy). There are three important studies, two listed below, that the authors need to include in their review and correctly discuss. The authors do reference one important ceramide/mitophagy study (citation#76, line 149, Sentelle et al.), but incorrectly associate the study with the citation that follows (citation#77). Citation #77 focuses on a role for c-jun in ceramide-mediated autophagy, but Sentelle et al (citation#76) does not. Sentelle et al. should be discussed alongside the two other studies from the Ogretmen group, which are mentioned below. These collectively discuss ceramide regulation of mitophagy, including a recent mechanism involving p17/PERMIT-mediated movement of CerS1 to mitochondria.

Oleinik N, Kim J, Roth BM, Selvam SP, Gooz M, Johnson RH, Lemasters JJ, Ogretmen B. Mitochondrial Protein Import Is Regulated by p17/PERMIT to Mediate Lipid Metabolism and Cellular Stress. Sci Adv. 2019 Sep 11;5(9):eaax1978. doi: 10.1126/sciadv.aax1978

Dany M, Gencer S, Nganga R, Thomas RJ, Oleinik N, Baron KD, Szulc ZM, Ruvolo P, Kornblau S, Andreeff M, Ogretmen B. Targeting FLT3-ITD Signaling Mediates Ceramide-Dependent Mitophagy and Attenuates Drug Resistance in AML. Blood. 2016 Oct 13;128(15):1944-1958. doi: 10.1182/blood-2016-04-708750

Response: Following the suggestion, we have described the interaction between ceramide and mitophagy using the studies by Sentelle et al (ref. 83) and Oleinik et al.(ref 84) (page 4, lines 157-161).

  1. The authors discuss ceramide regulation of the NADPH oxidase (NOX) in Section 3 (second paragraph, lines 113-123). However, this was incomplete and focused only on ceramide generated through the acid sphingomyelinase pathway in endothelial cells. The authors should also discuss the following study where the neutral sphingomyelinase has been shown to mediated NOX activity in neurons and neuroblastoma cells.

Barth BM, Gustafson SJ, Kuhn TB. Neutral Sphingomyelinase Activation Precedes NADPH Oxidase-Dependent Damage in Neurons Exposed to the Proinflammatory Cytokine Tumor Necrosis factor-α. J Neurosci Res. 2012 Jan;90(1):229-42. doi: 10.1002/jnr.22748

Response: A paper by Barth et al. (ref. 59) has been added (page 4, lines 126-128).

  1. The authors should also discuss the two studies listed below, which demonstrated that the ceramide-metabolite C1P can regulate NOX. This would be a logical progression from their discussion of ceramide-mediation of NOX (see comment #3 above), as well as logical given the author’s later discussion of CerK inhibitors.

Barth BM, Gustafson SJ, Hankins JL, Kaiser JM, Haakenson JK, Kester M, Kuhn TB. Ceramide Kinase Regulates TNFα-stimulated NADPH Oxidase Activity and Eicosanoid Biosynthesis in Neuroblastoma Cells. Cell Signal. 2012 Jun;24(6):1126-33. doi: 10.1016/j.cellsig.2011.12.020

Arana L, Gangoiti P, Ouro A, Rivera IG, Ordoñez M, Trueba M, Lankalapalli RS, Bittman R, Gomez-Muñoz A. Generation of Reactive Oxygen Species (ROS) Is a Key Factor for Stimulation of Macrophage Proliferation by Ceramide 1-phosphate. Exp Cell Res. 2012 Feb 15;318(4):350-60. doi: 10.1016/j.yexcr.2011.11.013

Response: Papers by Barth et al. (ref. 60) and Arana et al. (ref. 48) have been added (page 4, lines 126-128 and page 3, lines 98-99).

  1. Single-dose radiation therapy and the role of acid sphingomyelinase in its efficacy has been recently documented in a study that included a clinical trial. It is recommended that the authors include this study in their review in Section 4.

Bodo S, Campagne C, Thin TH, Higginson DS, Vargas HA, Hua G, Fuller JD, Ackerstaff E, Russell J, Zhang Z, Klingler S, Cho H, Kaag MG, Mazaheri Y, Rimner A, Manova-Todorova K, Epel B, Zatcky J, Cleary CR, Rao SS, Yamada Y, Zelefsky MJ, Halpern HJ, Koutcher JA, Cordon-Cardo C, Greco C, Haimovitz-Friedman A, Sala E, Powell SN, Kolesnick R, Fuks Z Single-dose Radiotherapy Disables Tumor Cell Homologous Recombination via ischemia/reperfusion Injury. .J Clin Invest. 2019 Feb 1;129(2):786-801. doi: 10.1172/JCI97631

Response: The single-dose radiation therapy has been described using the paper by Bodo et al. (ref 102)(page 5-6, lines 190-193,).

  1. FTY720 is discussed in Section 6.3.3 as a subsection for SphK inhibitors. This is somewhat misleading as it is not primary known as a SphK inhibitor. It is recommended that a separate section be created for S1PR regulators, which is what FTY720 is best recognized as and is its clinically-recognized/approved activity (for treating MS). A separate section for S1PR regulators would also allow the authors the opportunity to discuss other S1P regulators as mentioned below in comment #8. In addition, the authors should mention that FTY720 is now known as Fingolimod, which will help others more easily search for information about this drug

Response: We had used two references to state that FTY720 is an immunosuppressant for recurrent multiple sclerosis (page 8, lines 331-333). Fingolimod has been added (page 8, line 329).

  1. ABC294640 is discussed in Section 6.3.4. The authors should mention that ABC294640 is now known as Opaganib, which will help others more easily search for information about this drug. In addition, this section could be improved by mentioning that ABC294640 is currently in clinical trials for prostate cancer, cholangiocarcinoma (follow-up trials), as well as for COVID-19 (which presently is a very important topic).

Response: The word “Opaganib” has been added (page 9, line 353). We cited a phase 1 clinical trial for solid tumors (page 10, lines 363). The effect of ABC294640 on COVID-19 may be a topic, but we cannot find articles confirming its effect in the literature.

  1. The authors missed discussing some prominent inhibitors/regulators. These mostly are inhibitors or regulators of CDase (SACLAC), SphK (MP-A08, SK1-I), S1PR (CYM-5478), and GluCS/P-Gp (tamoxifen, zosuquidar). Some of the relevant literature that should be included is listed as follows.

Pearson JM, Tan SF, Sharma A, Annageldiyev C, Fox TE, Abad JL, Fabrias G, Desai D, Amin S, Wang HG, Cabot MC, Claxton DF, Kester M, Feith DJ, Loughran TP.Ceramide Analogue SACLAC Modulates Sphingolipid Levels and MCL-1 Splicing to Induce Apoptosis in Acute Myeloid Leukemia. Mol Cancer Res. 2020 Mar;18(3):352-363. doi: 10.1158/1541-7786.MCR-19-0619

Pitman MR, Powell JA, Coolen C, Moretti PA, Zebol JR, Pham DH, Finnie JW, Don AS, Ebert LM, Bonder CS, Gliddon BL, Pitson SM.A Selective ATP-competitive Sphingosine Kinase Inhibitor Demonstrates Anti-Cancer Properties. Oncotarget. 2015 Mar 30;6(9):7065-83. doi: 10.18632/oncotarget.3178

Powell JA, Lewis AC, Zhu W, Toubia J, Pitman MR, Wallington-Beddoe CT, Moretti PA, Iarossi D, Samaraweera SE, Cummings N, Ramshaw HS, Thomas D, Wei AH, Lopez AF, D'Andrea RJ, Lewis ID, Pitson SM.Targeting Sphingosine Kinase 1 Induces MCL1-dependent Cell Death in Acute Myeloid Leukemia. Blood. 2017 Feb 9;129(6):771-782. doi: 10.1182/blood-2016-06-720433

Pitman MR, Costabile M, Pitson SM.Recent Advances in the Development of Sphingosine Kinase Inhibitors. Cell Signal. 2016 Sep;28(9):1349-63. doi: 10.1016/j.cellsig.2016.06.007

Edmonds Y, Milstien S, Spiegel S.Development of Small-Molecule Inhibitors of sphingosine-1-phosphate Signaling. Pharmacol Ther. 2011 Dec;132(3):352-60. doi: 10.1016/j.pharmthera.2011.08.004

Wang W, Shanmugam MK, Xiang P, Yam TYA, Kumar V, Chew WS, Chang JK, Ali MZB, Reolo MJY, Peh YX, Karim SNBA, Tan AYY, Sanda T, Sethi G, Herr DR.Sphingosine 1-Phosphate Receptor 2 Induces Otoprotective Responses to Cisplatin Treatment. Cancers (Basel). 2020 Jan 15;12(1):211. doi: 10.3390/cancers12010211

Wang W, Xiang P, Chew WS, Torta F, Bandla A, Lopez V, Seow WL, Lam BWS, Chang JK, Wong P, Chayaburakul K, Ong WY, Wenk MR, Sundar R, Herr DR.Activation of Sphingosine 1-phosphate Receptor 2 Attenuates Chemotherapy-Induced Neuropathy. J Biol Chem. 2020 Jan 24;295(4):1143-1152. doi: 10.1074/jbc.RA119.011699

Morad SAF, Davis TS, MacDougall MR, Tan SF, Feith DJ, Desai DH, Amin SG, Kester M, Loughran TP Jr, Cabot MC.Role of P-glycoprotein Inhibitors in Ceramide-Based Therapeutics for Treatment of Cancer. Biochem Pharmacol. 2017 Apr 15;130:21-33. doi: 10.1016/j.bcp.2017.02.002

Response: We have added the papers of Pearson et al. (ref 123), Pitmen et al. 2015 (ref.131), Powell et al. (ref.133), and Edmonds et al. (ref. 130)(page 7, lines 260-261). The content of the paper of Morad et al. (ref. 128) has been described (page 7, lines 251-254). The papers by Wang et al. are not included, because the authors evaluated the cytoprotective effects of the S1P2 agonist CYM-5478 on cisplatin-induced adverse events.

  1. The authors do a reasonable job discussing ceramide-technologies in Section 8, which includes a modest but incomplete discussion about the ceramide nanoliposome. Given the focus and intent of their review, the authors should indicate that this technology is in clinical trials for cancer (Ceramide NanoLiposome: ClinicalTrials.gov Identifier: NCT02834611). The authors list some of the cancers that the ceramide nanoliposome has been shown to treat, but this is an incomplete list and the referencing is very minimal. More so, the authors only elaborate on one study (citation #184, Jiang et al.) where the ceramide nanoliposome was tested in combination with chemotherapy. There are many other studies evaluating the combination of the ceramide nanoliposome with chemotherapy. The authors should add the following studies, which are the most recent studies (acute myeloid leukemia, colorectal cancer, liver cancer immune responses, ovarian cancer) as well as a review of its preclinical development.

Barth BM, Wang W, Toran PT, Fox TE, Annageldiyev C, Ondrasik RM, Keasey NR, Brown TJ, Devine VG, Sullivan EC, Cote AL, Papakotsi V, Tan SF, Shanmugavelandy SS, Deering TG, Needle DB, Stern ST, Zhu J, Liao J, Viny AD, Feith DJ, Levine RL, Wang HG, Loughran TP Jr, Sharma A, Kester M, Claxton DF. Sphingolipid Metabolism Determines the Therapeutic Efficacy of Nanoliposomal Ceramide in Acute Myeloid Leukemia. Blood Adv. 2019 Sep 10;3(17):2598-2603. doi: 10.1182/bloodadvances.2018021295

Lu P, White-Gilbertson S, Nganga R, Kester M, Voelkel-Johnson C. Expression of the SNAI2 Transcriptional Repressor Is Regulated by C 16-ceramide. Cancer Biol Ther. 2019;20(6):922-930. doi: 10.1080/15384047.2019.1579962

Li G, Liu D, Kimchi ET, Kaifi JT, Qi X, Manjunath Y, Liu X, Deering T, Avella DM, Fox T, Rockey DC, Schell TD, Kester M, Staveley-O'Carroll KF. Nanoliposome C6-Ceramide Increases the Anti-tumor Immune Response and Slows Growth of Liver Tumors in Mice. Gastroenterology. 2018 Mar;154(4):1024-1036.e9. doi: 10.1053/j.gastro.2017.10.050

Zhang X, Kitatani K, Toyoshima M, Ishibashi M, Usui T, Minato J, Egiz M, Shigeta S, Fox T, Deering T,Kester M, Yaegashi N. Ceramide Nanoliposomes as a MLKL-Dependent, Necroptosis-Inducing, Chemotherapeutic Reagent in Ovarian Cancer. Mol Cancer Ther. 2018 Jan;17(1):50-59. doi: 10.1158/1535-7163.MCT-17-0173

Kester M, Bassler J, Fox TE, Carter CJ, Davidson JA, Parette MR. Preclinical Development of a C6-ceramide NanoLiposome, a Novel Sphingolipid Therapeutic. Biol Chem. 2015 Jun;396(6-7):737-47. doi: 10.1515/hsz-2015-0129

Response: We have added the papers of Li et al (ref. 199), Zhang (ref. 200), and Kester (ref.198), that studied the therapeutic effect on solid tumors.

  1. There are several minor errors as follows.

Response: Thank you for indicating the following errors.

Page 2, line 84: spelling error “shingosine-1-phosphate”

Response: “shingosine-1-phosphate” has been changed to “sphingosine-1-phosphate” (page 2, line 85).  

Page 2, line 92: spelling error “sphigomyelin synthase”

Response: “sphigomyelin synthase” has been changed to “sphingomyelin synthase” (page 2, line 93).

Page 6, line 197: spelling error “CesS1”

Response: “CesS1” has been changed to “CerS1” (page 6, line 212).

Page 9, line 351: grammar error “has”

Response: “has” has been deleted.(page 10, line 373).

Figure 1: spelling error “3-ketosphinganie reductase”; “Dihydrosphindosine”

Response: “3-ketosphinganie reductase”; “Dihydrosphindosine” have been changed to “3-ketosphinganine reductase”; “Dihydrosphingosine”

Figure 2: spelling error “Cytocrome”

Response: “Cytocrome” has been changed to “Cytochrome”

Figure 3A: spelling error “sphigosine”

Response: “sphigosine” has been changed to “sphingosine”

Special (Greek) characters are omitted throughout the manuscript.

Response: This may occur during the conversion of font. We have added the characters.

Please double-check the manuscript for other possible spelling/grammar errors

Response: We have rectified other errors found in this time.

  1. The authors cite a substantial number of review articles rather than the primary literature. For reviews, it is usually best to cite the original research when possible. However, it is understandable that some reviews will ultimately be cited. This might be appropriate in some cases, especially when the reviews are prominent and/or summarizing a large body of work.

  1. The focus of this review according to the title and abstract is on oral squamous cell carcinoma (SCC). However, much of the manuscript summarizes work that was not conducted on SCC. This is a minor concern that the authors might address in their abstract. It could be highlighted that much of the reviewed material, even if not specifically using SCC models, is directly-applicable to SCC as the fundamental target is ceramide metabolism.

Response: We have changed “oral squamous cell carcinoma” to “head and neck squamous cell carcinoma” in the title.

Round 2

Reviewer 2 Report

The authors have only made minor alterations to the manuscript. Significant concerns still remain. Specifically , as mentioned in my previous review

However, it isn’t clear whether the amount of data available for OSCC (or HNSCC) is sufficient to focus entirely on this disease, as the relevant data are intertwined with information from other tumour types throughout the manuscript, leaving the reader unsure of the potential of these inhibitors as therapeutic agents for OSCC (HNSCC). If there are sufficient data available related specifically to OSCC (or HNSCC), the review would need to be refocused and reorganized to be suitable for publication.

Similarly, in some other sections there is little or no mention of OSCC (or HNSCC). The authors should consider an entirely separate section on OSCC (HNSCC) and perhaps describe what is known about the sphingolipid pathway in OSCC (HNSCC), such as ceramide levels, SPHK1 expression etc and how these might be targeted with inhibitors.

These issues have not been addressed in any way. The authors have simply changed OSCC to HNSCC (as I suggested), but this is rather a point of detail and doesn't address the above concerns. 

Author Response

Comments and Suggestions for Authors

The authors have only made minor alterations to the manuscript. Significant concerns still remain. Specifically , as mentioned in my previous review

However, it isn’t clear whether the amount of data available for OSCC (or HNSCC) is sufficient to focus entirely on this disease, as the relevant data are intertwined with information from other tumour types throughout the manuscript, leaving the reader unsure of the potential of these inhibitors as therapeutic agents for OSCC (HNSCC). If there are sufficient data available related specifically to OSCC (or HNSCC), the review would need to be refocused and reorganized to be suitable for publication.

Response: Thank you for your valuable comments. We have re-investigated HNSCC-related articles, particularly clinical and preclinical studies. We focused on the literature examining the effects of inhibitors as potential agents for treating HNSCC and reaffirmed that the information related with HNSCC is scarce. Most papers are already included in this review paper, but as the reviewer has shown, these papers are embedded in information performed in other types of carcinoma. Two preclinical and clinical studies have been added to the revised version to increase the amount of information about HNSCC.The papers are 1) Zhao et al. MiR-124 acts as a tumor suppressor by inhibiting the expression of sphingosine kinase 1 and its downstream signaling in head and neck squamous cell carcinoma. Oncotarget. 2017, 8, 25005 (ref. 131) and 2) Kunii, et al. Combination therapy of in vitro-expanded natural killer T cells and alpha-galactosylceramide-pulsed antigen-presenting cells in patients with recurrent head and neck carcinoma. Cancer Sci. 2009,100,1092 (ref. 215).The following sentences have been added to show the involvement of HNSCC in each section. Section 5: gemcitabine and doxorubicin combination reconstituted levels of C18-ceramide vis up-regulation of CerS1 expression in HNSCC (page 6, lines 210-215). Section 6.3: The role of SphK in HNSCC (page 7, lines 265-271).

The following sentences have been added to show the involvement of HNSCC in each section. Section 5: gemcitabine and doxorubicin combination reconstituted levels of C18-ceramide vis up-regulation of CerS1 expression in HNSCC (page 6, lines 210-215). Section 6.3: The role of SphK in HNSCC (page 7, lines 265-271).

Similarly, in some other sections there is little or no mention of OSCC (or HNSCC). The authors should consider an entirely separate section on OSCC (HNSCC) and perhaps describe what is known about the sphingolipid pathway in OSCC (HNSCC), such as ceramide levels, SPHK1 expression etc and how these might be targeted with inhibitors.

 These issues have not been addressed in any way. The authors have simply changed OSCC to HNSCC (as I suggested), but this is rather a point of detail and doesn't address the above concerns. 

Response: Following the suggestion of the reviewer, we have added a separate section titled “10. Application of sphingolipid target therapy to HNSCC”. In that section, the numbers of sections describing events such as ceramide levels and SphK1 expression in HNSCC are shown in parentheses (pages 12, lines 461-489).  

This time, we have changed “head and neck SCC” to “HNSCC” in the text.

Round 3

Reviewer 2 Report

The manuscript has been improved. The additional section on HNSCC is useful and improves the article.